# Body composition among Malawian young adolescents: Cross-validating predictive equations for bioelectric impedance analysis using deuterium dilution method

Pieta Näsänen-Gilmore[1,2], Chiza Kumwenda[3☯], Markku Nurhonen[2☯], Lotta Hallamaa[1], Charles Mangani[4], Per Ashorn[1,5], Ulla Ashorn[1], Eero Kajantie[2,6,7,8]*

1 Center for Child, Adolescent, and Maternal Health Research, Faculty of Medicine and Health Technology, Tampere University, Tampere, Finland, 2 Population Health Unit, Finnish Institute for Health and Welfare, Helsinki and Oulu, Finland, 3 Department of Food Science and Nutrition, University of Zambia, Lusaka, Zambia, 4 Department of Public Health, School of Public Health and Family Medicine, Kamuzu University of Health Sciences, Blantyre, Malawi, 5 Department of Paediatrics, Tampere University Hospital, Tampere, Finland, 6 Department of Clinical and Molecular Medicine, Norwegian University of Science and Technology, Trondheim, Norway, 7 Clinical Medicine Research Unit, MRC Oulu, Oulu University Hospital and University of Oulu, Oulu, Finland, 8 Children's Hospital, Helsinki University Hospital and University of Helsinki, Helsinki, Finland

☯ These authors contributed equally to this work.
* eero.kajantie@thl.fi

## Abstract

### Background

Body composition can be measured by several methods, each with specific benefits and disadvantages. Bioelectric impedance offers a favorable balance between accuracy, cost and ease of measurement in a range of settings. In this method, bioelectric measurements are converted to body composition measurements by prediction equations specific to age, population and bioimpedance device. Few prediction equations exist for populations in low-resource settings. We formed a prediction equation for total body water in Malawian adolescents using deuterium dilution as reference.

### Methods

We studied 86 boys and 92 girls participating in the 11-14-year follow-up of the Lungwena Antenatal Intervention Study, a randomized trial of presumptive infection treatment among pregnant women. We measured body composition by Seca m515 bioimpedance analyser. Participants ingested a weight-standardized dose of deuterium oxide, after which we collected saliva at baseline, at 3 and 4 h post-ingestion, measured deuterium concentration using Fourier-transform infrared spectroscopy and calculated total body water. We formed predictive equations for total body water using anthropometrics plus resistance and reactance at a range of frequencies, applying multiple regression and repeated cross-validation in model building and in prediction error estimation.

**Data Availability Statement:** Data are sensitive health data and cannot be shared publicly. Data requests from researchers who meet the criteria for access to confidential information can be made to Data Manager Juho Luoma (juho.luoma@tuni.fi).

**Funding:** The original LAIS study was supported by grants from the Academy of Finland (grants 79787 and 207010 to PA; aka.fi), the Foundation of Pediatric Research in Finland (lastentautientutkimussaatio.fi; PA), and the Medical Research Fund of Tampere University Hospital (PA). This study was supported by the Academy of Finland (EK; aka.fi), Finnish Medical Foundation (Finska Läkaresällskapet) (EK), Foundation for Pediatric Research in Finland (EK, lastentautientutkimussaatio.fi), Novo Nordisk Foundation (EK, novonordiskfonden.dk), Signe and Ane Gyllenberg Foundation (PNG, EK; gyllenbergs.fi) and Sigrid Juselius Foundation (EK; sigridjuselius.fi). The funders had no role in study design, data collection and analysis, decision to publish, or preparation of the manuscript.

**Competing interests:** The authors have declared that no competing interests exist.

## Results

The best predictive model for percentage total body water (TBW %) was $100*(1.11373 + 0.0037049*\text{height (cm)}^2/\text{resistance}(\Omega)$ at 50 kHz $- 0.25778*\text{height(m)} - 0.01812*\text{BMI(kg/m}^2) - 0.02614*\text{female sex})$. Calculation of absolute TBW (kg) by multiplying TBW (%) with body weight had better predictive power than a model directly constructed to predict absolute total body water (kg). This model explained 96.4% of variance in TBW (kg) and had a mean prediction error of 0.691 kg. Mean bias was 0.01 kg (95% limits of agreement -1.34, 1.36) for boys and -0.01 kg (1.41, 1.38) for girls.

## Conclusions

Our equation provides an accurate, cost-effective and participant-friendly body composition prediction method among adolescents in clinic-based field studies in rural Africa, where electricity is available.

## Introduction

Body composition is a sensitive indicator of nutritional status [1]. Body mass index (BMI), which applies basic anthropometric measurements (weight and height), has been widely used in determining body composition and nutritional status due to its simplicity and applicability [2]. However, BMI does not distinguish between fat and lean tissue, which is problematic in particular in children and adolescents due to high variablilty in body proportions [2–4]. Mean BMI is currently increasing in most populations, including populations where undernutrition due to inadequate energy intake has been prevalent [5]. These populations, mostly in low resource settings, are increasingly suffering from a combination of obesity and inadequate intake of essential nutrients. Understanding the body composition and its change among youth in these settings is crucial for improving nutritional status and health [6]. This requires an accurate, simple, affordable and easy-to-administer method to assess body composition.

Bioelectric impedance analysis (BIA) is a fairly simple method used to estimate body composition. Balancing accuracy, reproducibility, and ease of measure, BIA is likely the preferred method in field-based population studies [7]. In this method, small electrodes are used to direct a weak alternating current through the body to measure resistance and reactance, the two components of impedance [8]. Resistance and reactance depend, among other determinants, on water content of the tissue and, with appropriate prediction equations, can therefore be used to calculate total body water (TBW). TBW is then used to calculate fat-free mass (FFM) based on the relatively constant FFM water content of 73% in healthy adults and 75–76% in children [9–12]. Fat mass (FM) is then calculated by subtracting FFM from total body weight.

Deuterium dilution method serves as gold standard for measuring TBW [8, 13]. However, this method is labour-intensive, lengthy to perform and relatively costly [13] and does not allow free intake of liquid during the procedure, and hence requires some effort to carry out on children or young adolescents. BIA method would offer an excellent simple alternative for body composition analysis, but whenever a new type of device is used or a new population studied, new prediction equations will first need to be created and validated [8, 14]. Few validation studies using deuterium dilution as reference have been published for children in populations in Sub-Saharan Africa, and all of those we are aware of have been conducted in West

Africa [15–17]. Moreover, one study compared different prediction equations in preadolescent South African children using dual x-ray absorptiometry as reference [18].

We carried out a cross validation study to create and validate prediction equations for body composition measurements by Seca m515 8-polar Bioimpedance analyser for adolescents aged 11–14 years of age in Lungwena, Malawi, using deuterium dilution techniques as a reference technique.

## Materials and methods

### Ethical considerations

The study protocol was approved by the College of Medicine Research and Ethics Committee, Blantyre, Malawi. The Lungwena Antenatal Intervention Study (LAIS) protocol has also been approved by the Ethics Committee of Pirkanmaa Hospital District, Finland. We obtained an assent from each participant and a written informed consent from their guardian on the day of the testing. The consent form was read to the guardian in their preferred language (chewa or yao) and any questions were answered before the guardian signed the consent form by a signature or a thumbprint. For more detail on community involvement, see S1 File.

We carried out a study to validate the bioelectric impedance method by Seca mBCA 515 8-polar bioimpedance (Seca, Germany) using the gold standard deuterium dilution technique as a reference method [13] for the use of body composition analysis of children and young adolescents aged 11–14 years in Malawi. This study ran from September to December in 2018 within the already existing framework of LAIS data collection that took place between December 2017 and March 2019 [19]. LAIS is a longitudinal follow-up cohort of children whose mothers participated in a randomized, controlled trial of presumptive infection treatment during their pregnancy in Lungwena, Southern Malawi [20, 21].

In order to attain participation by approximately 100 boys and 100 girls, we used random sequence generator method (random.org) to randomly select 120 boys and 120 girls (total of n = 240) to be invited to the study (Fig 1). The selection was made among the 728 LAIS study participants who had attended the data collection by 12 September 2018. There were no exclusion criteria. 96 boys and 104 girls attended the study (Fig 1).

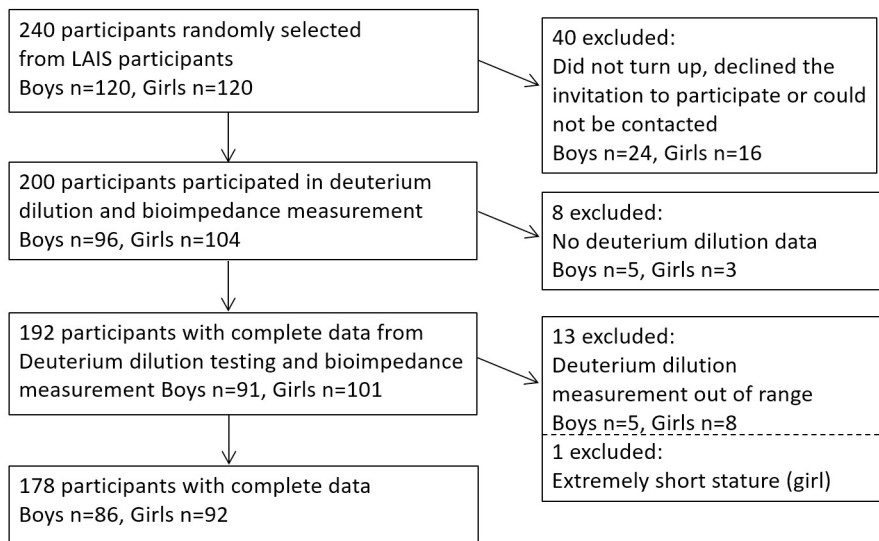

**Fig 1. Flow chart.** LAIS, Lungwena Antenatal Intervention Study.

Prior to the study visit day, study coordinators visited participant households to invite the selected participants for this validation study, and discussed the purpose and procedures of the study. Participants were instructed to fast overnight for 6 hours minimum with no ingestion of food or drink after going to sleep at night or in the morning. All participants and their accompanying parent/guardian were asked to attend the clinic in the morning.

## Study day

On arrival study coordinators further explained the purpose of the study, and the schedule for the day, after which the coordinator sought the informed assents and consents from all participants and their guardians. After this, each participant was weighed using the Seca m515 bioimpedance machine (Seca, Germany). Two anthropometrists measured height with a Harpenden stadiometer (Holtain Ltd, UK) and waist circumference, under the lowest rib, using Seca Tape measure. Study nurse assessed the pubertal stage using Tanner's score for pubic hair development and interviewed participants about the general health status and whether or not they had fasted overnight. If the participant had not fasted, they were asked to report the hours of fasting prior to arriving at the clinic and to estimate liquid intake in ml, and/or food intake. The participants also reported on the mode of transport to the visit, which served as a proxy for physical activity prior to the testing. They also reported whether or not they had emptied the bladder just at the start of the day. After the interview and anthropometrics, the participants underwent body composition measurement by bioimpedance and the gold standard method of deuterium dilution.

## Body composition by bioimpedance

We carried out bioimpedance analysis using Seca m515 8-polar bioimpedance analyser (Seca, Germany) which measured impedance and its components resistance and reactance, and phase angle for 19 frequencies ranging from 1 kHz to 1000 kHz. All raw data were collected for each participant. The participants wore light clothing, and 0.5 kg was automatically subtracted from their weight. They stood on the machine barefoot such that their heels are were placed on the back electrodes and the front of their feet are on the front electrodes. They placed their hands on the hand electrodes such that their arms were extended but not strained and that each side of an electrode held two fingers.

## Body composition by deuterium dilution

After bioimpedance measurement, the participants underwent deuterium dilution assessment following the guidelines of the International Atomic Energy Agency (IAEA) [13]. Deuterium dioxide is equally distributed in all liquid body water compartments including saliva [13]. First the participants provided the baseline saliva sample. They were given sterile cotton wool to chew until wet soaked in saliva (approximately for 5 minutes), after which the cotton was spat in a sterile syringe from which saliva was squirted out into a 2 ml tube (Thermo Scientific, Finland). Minimum requirement was 1 ml of saliva. If less was obtained, the participant was asked to chew another cotton wool in order to obtain more saliva. A back up sample was also collected so the general aim was to collect 2 x 1ml of saliva from each participant. Participants were treated as a group of 5 individuals at a time in order to ease the timing of dosing and handling of saliva sampling.

The participants then received a dose of deuterium dioxide mixed in sterile water. The dose was dependent on the weight of the participant: a dose of 10 ml was given to those weighing less than 30 kg: dose of 20 ml to those weighing 30 to 50 kg: or 30 ml to those weighing over 50 kg, following the IAEA protocol [13]. Deuterium dioxide dose was offered from a sterile dose

bottle prepared the day before testing. The bottle was labelled with a unique identifier number which was recorded on the clinic form and was used as a key to link participant to clinical data. Dose bottles containing a specific weight were topped up to 50 ml of sterile water using a 100ml sterile cylinder and participants were asked to drink the liquid to ingest the whole deuterium dose. The bottle was again filled up to 50ml to wash off the rest of the dose. Study coordinators recorded the time of dosing on the clinic visit form. The participants were instructed not to drink or pass any urine unless necessary until the 3 hour and 4 hour post-dose saliva samplings. Saliva was collected at 3 and 4 hours after ingestion of deuterium using the method described above. At completion of saliva samplings participants were weighed for the final time using the bioimpedance machine (final weight) and any drinking any liquid or passing urine during the visit day were recorded. At the end of the study day participants were offered a bottle of soda, a small snack as well as an incentive, which was a kilo of flour and a bar of soap.

## Analysis of deuterium oxide concentrations

Saliva samples from the field were carried back to the Public Health Nutrition Laboratory at the College of Medicine in Mangochi, Malawi in cool boxes filled with frozen ice packs. The journey took about 45 minutes by car. The baseline and postdose saliva samples for each participant were kept separate in plastic ziplock bags to avoid cross contamination, to prevent evaporation from the samples and to avoid environmental moisture exposure. Saliva samples were stored at -20 C awaiting analysis. Fourier transform infrared spectroscopy (Agilent 4500; Agilent Technologies) was used to analyse samples for deuterium enrichment. Frozen saliva samples were thawed at room temperature before analysis. The baseline saliva sample was used as the background for each of the participant's postdose samples and enrichment was recorded in parts per million (ppm). Quality control procedures outlined in the IAEA manual were applied [13]. The precision of the FTIR was assessed on a daily basis for three days. Within and between-day coefficients of variation for the deuterium standard were ≤0.55% which is within the range (<1%) recommended by the IAEA [13].

## Total body water

We calculated total body water using saliva deuterium concentration following IAEA methods [13]. Five boys and eight girls with missing or implausible (over 80% or under 35%) TBW (%) values and one very short girl (1.1 m; -4.4 SD) were excluded (Fig 1). We applied age specific hydration factors 0.754 and 0.747 (boys) and 0.766 and 0.755 (girls) in age groups 11–12 and 13–14 years, respectively [13, 22]. Fat free mass was obtained by dividing TBW (kg) by this value and fat mass by subtracting fat free mass from body weight.

## Statistical analysis

We used data from 86 boys and 92 girls. A total of 23 variables were considered for inclusion in the prediction equations. Variables that we applied in the modelling procedure were sex, age (in years), height (m), weight (kg), BMI, waist circumference (m), weight loss during the study day (kg), Tanner scale (1–5) for stage of puberty, reactance at 50KHz and resistance at 5 frequencies as listed in Table 1, and three dichotomous variables: mode of transport (walking or cycling = 0; bus, car, motorbike or bicycle taxi = 1), fasting overnight (no = 0; yes = 1), bladder emptied in the morning (no = 0; yes = 1). For these dichotomous variables, missing values (Table 1) were coded as 0.5. No other variables had missing values. Resistance index (also referred to as impedance index), which in theory is directly proportional with TBW [9], was obtained by dividing squared height (cm) by resistance, and reactance indices accordingly. Of

**Table 1. Anthropometric and other characteristics applied in the modelling procedure.**

| | Boys (n = 86) | | Girls (n = 92) | |
|---|---|---|---|---|
| | Mean | SD | Mean | SD |
| Age (y) | 13.0 | 0.8 | 12.9 | 0.8 |
| Height (cm)[a] | 141 | 8 | 146 | 8 |
| Height z-score[a] | -2.05 | 0.85 | -1.30 | 0.95 |
| Stunted, n (%)[a] | 44 | (51.2%) | 21 | (22.8%) |
| Weight (kg) | 31.9 | 5.5 | 36.1 | 6.9 |
| BMI (kg/m$^2$)[b] | 16.0 | 1.4 | 16.7 | 1.9 |
| BMI z-score[b] | -1.35 | 0.87 | -1.03 | 0.90 |
| BMI z-score <-2, n (%)[b] | 26 | (30.2%) | 12 | (13.0%) |
| Waist circumference (cm) | 61.7 | 4.2 | 63.6 | 5.0 |
| Tanner pubic hair stage, n (%) | | | | |
| 1 | 78 | (90.7%) | 26 | (28.3%) |
| 2 | 6 | (7.0%) | 34 | (37.0%) |
| 3 | 2 | (2.3%) | 19 | (20.7%) |
| 4 | 0 | | 13 | (14.1%) |
| 5 | 0 | | 0 | |
| Weight loss during study day (kg) | 0.31 | 0.21 | 0.32 | 0.24 |
| Arrived by transport, n (%)[c] | 6 | (7.1%) | 16 | (17.6%) |
| Fasted overnight, n (%)[d] | 38 | (44.7%) | 52 | (56.5%) |
| Bladder emptied in the morning, n (%)[e] | 46 | (58.2%) | 50 | (58.8%) |
| Resistance at 50 kHz (Ω) | 440 | 56 | 419 | 44 |
| Reactance at 1 kHz (Ω) | 6.11 | 1.19 | 6.44 | 1.2 |
| Reactance at 15 kHz (Ω) | 24.3 | 3.5 | 24.9 | 3.6 |
| Reactance at 50 kHz (Ω) | 35.9 | 4.8 | 35.0 | 4.3 |
| Reactance at 200 kHz (Ω) | 38.1 | 5.7 | 35.4 | 4.6 |
| Reactance at 1000 kHz (Ω) | 44.2 | 11.3 | 40.0 | 8.4 |
| Resistance index at 50 kHz (cm$^2$/Ω) | 46.2 | 10.0 | 52.2 | 9.4 |
| Reactance index at 1 kHz (cm$^2$/Ω) | 3391 | 829 | 3485 | 886 |
| Reactance index at 15 kHz (cm$^2$/Ω) | 837 | 161 | 886 | 187 |
| Reactance index at 50 kHz (cm$^2$/Ω) | 566 | 113 | 629 | 126 |
| Reactance index at 200 kHz (cm$^2$/Ω) | 538 | 127 | 623 | 131 |
| Reactance index at 1000 kHz (cm$^2$/Ω) | 490 | 176 | 569 | 160 |

[a]Only height (cm) was applied in modelling. Stunted = height z-score <-2 SD.

[b]Only BMI (kg/m$^2$) was applied in modelling.

[c]Missing for 1 boy and 1 girl

[d]Missing for 1 boy

[e]Missing for 7 boys and 7 girls

note, the original data consisted of reactance and resistance both measured at 19 frequencies ranging from 1 to 1000 kHz. Due to strong multicollinearity, we limited the number of bioimpedance variables in modelling. This eased model building without affecting the predictive performance of the resulting equations. Impedance is a complex number and the vector sum of resistance (real part) and reactance (imaginary part) [23]. All resistance values were heavily correlated (pairwise correlations 0.98–1.00), and only resistance at 50 kHz was used. Reactance varies by frequency of the sinusoidal alternating current [23]. Accordingly, reactance measurements at separate frequencies were dissimilar and values at five frequencies, excluding mutually correlated variables, were applied in the modelling procedure.

Prediction equations were obtained using multiple linear regression models by applying three alternative variable selection methods. Using R package leaps [24, 25], we utilised the best subsets selection algorithm and identified models with minimum Bayesian (BIC) and Akaike (AIC) information criterion values. As a third method, using R package caret [26], we applied repeated 10-fold cross validation (RCV) to determine the optimal number of variables [27–29]. We applied the backward variable selection algorithm and, in RCV, the one standard error rule, thus favoring more parsimonious models.

To assess out-of-sample (external) performance, we set aside one tenth of the data as a test set, used the rest of the data to form predictive equations and then applied the equations on the test set. For this purpose, the data was randomly split into 10 folds and each fold in turn was used as a test set, thus producing an out-of-sample prediction for all observations. The corresponding squared prediction errors (PE) were combined to form the root mean squared error of predictions RMSE (PE), which was used to estimate the external prediction performance and to compare the variable selection methods. For increased accuracy in prediction error estimation, we calculated the average RMSE (PE)-value from 100 sets of random 10-fold splits for each of the three criteria (AIC, BIC and RCV). Importantly, the RMSE found for the optimal model during the model selection stage was not used in error estimation as this would imply using the same data for modeling and performance evaluation, leading to overly optimistic error estimates [30]. Instead, an out-of-sample prediction error estimate was obtained by predicting data that was not utilised in modeling. For RCV, this is the nested cross validation approach recommended e.g., by Krstajic et al [30]. To be utilised in scatterplots, we selected the set of predicted values whose RMSE (PE) was closest to the average RMSE (PE).

We considered four model types: joint and separate models by sex and prediction of the outcome TBW (kg) either directly or via the equation obtained for TBW (%). Best model type and variable selection method were determined by expected out-of-sample performance as measured by average RMSE (PE). The final equation was obtained as the multiple linear regression model applying the best variable selection method to the whole data. We also express the concordance between the predictive equation and deuterium dilution by using Bland-Altman plots [31] with 95% limits of agreement (± 1.96*SD).

The R code for cross-validating the predictive equations is available as S2 File.

## Results

Anthropometric and other characteristics applied in the modelling procedure are shown in Table 1. The ages of the children ranged from 11.5 to 14.5 years. Mean height z-scores of the boys and girls were -2.05 and -1.30, and 51.2% and 22.8% were stunted. Mean BMI z-scores were -1.35 and -1.03. Body composition measured by deuterium dilution is shown in Table 2. Scatterplots illustrating associations of anthropometric and bioimpedance measurements with TBW (kg) and TBW (%) measured by deuterium dilution are shown in S1 and S2 Figs.

**Table 2. Body composition measured by deuterium dilution.**

|  | Boys (n = 86) | | | | Girls (n = 92) | | | |
|---|---|---|---|---|---|---|---|---|
|  | **Mean** | **SD** | **Min** | **Max** | **Mean** | **SD** | **Min** | **Max** |
| Weight (kg) | 31.9 | 5.5 | 22.1 | 51.5 | 36.1 | 6.9 | 23.6 | 53.7 |
| Total body water (kg) | 20.2 | 3.6 | 14.4 | 32.0 | 21.6 | 3.6 | 14.8 | 30.6 |
| Total body water (%) | 63.3 | 2.5 | 57.4 | 69.9 | 60.1 | 3.3 | 50.5 | 66.7 |
| Fat mass (kg) | 5.0 | 1.4 | 2.1 | 8.9 | 7.7 | 2.7 | 3.3 | 17.0 |
| Fat mass (%) | 15.7 | 3.3 | 7.2 | 23.8 | 21.0 | 4.2 | 11.6 | 34.1 |
| Fat free mass (kg) | 26.9 | 4.8 | 19.1 | 42.9 | 28.4 | 4.8 | 19.3 | 40.6 |

## Predictive equations for total body water

To choose an optimal variable selection criterion, we first calculated predictive equations for TBW (kg) and TBW (%) using the three variable selection criteria (BIC, AIC, RCV) and assessed their expected out-of-sample performance using the corresponding average RMSE (PE) values obtained from 100 repeated splits of data into 10 folds. Equations involving relatively few variables performed best. For example, when predicting TBW (kg) using a common model for both sexes, the average RMSE (PE) values corresponding to the best equation selected by BIC, AIC and RCV variable selection methods were 0.708 kg, 0.720 kg and 0.691 kg, respectively. RCV selected 4, BIC 5 and AIC 10 predictors. Therefore, we only present equations and estimated prediction errors obtained using RCV.

Table 3 shows comparisons of model performance with different modelling approaches to estimate TBW (kg) and TBW (%). We first compared joint models including both sexes with sex-stratified models. In each case the prediction error RMSE (PE), the root mean squared error of the out-of-sample predicted values, was smaller in a joint model for both sexes. We then compared two approaches to predict TBW (kg). Predicting TBW (kg) by multiplying predictions for TBW (%) by body weight had a somewhat better out-of-sample predictive performance than forming an equation directly predicting TBW (kg) (average prediction error 0.691 kg vs 0.734 kg) and it is therefore the preferred predictive model for TBW (kg).

The best performing prediction equations are shown in Table 4, which also shows, for comparison, the equation directly predicting TBW (kg).

The out-of-sample predicted values explain 96.4% of the variance of TBW (kg). For TBW (%), mean prediction error is 1.99% (percentage points), and the out-of-sample predicted

**Table 3. Predictive equations for total body water (TBW) in kg and TBW as a percentage of weight.** For each outcome, out-of-sample performance metrics (the last two columns) are given for the combined data and for boys and girls separately. Preditive models were fitted for both sexes together (sex specific intercept only) and separately for each sex (sex specific equations).

| Outcome | Model | Group | Variables | $r^2$ | Max (p) | RMSE (PE) (kg or %p) | $r^2$ (PE) |
|---|---|---|---|---|---|---|---|
| (a) Total body water (kg) | Both sexes | All | R, RI, wt, sex | 0.962 | 2e-05 | 0.734 | 0.959 |
| | | Boys | | | | 0.721 | 0.958 |
| | | Girls | | | | 0.747 | 0.956 |
| | Sex-specific | Boys | R, RI, wt | 0.965 | 0.004 | 0.726 | 0.958 |
| | | Girls | RI, wt, BMI | 0.963 | 0.001 | 0.761 | 0.955 |
| (b) Total body water (%) | Both sexes | All | RI, ht, BMI, sex | 0.660 | 3e-11 | 1.99%p* | 0.638 |
| | | Boys | | | | 2.12%p | 0.257 |
| | | Girls | | | | 1.85%p | 0.675 |
| | Sex-specific | Boys | RI, ht, BMI | 0.303 | 9e-04 | 2.29%p | 0.130 |
| | | Girls | RI, HT, BMI | 0.699 | 2e-08 | 1.93%p | 0.649 |
| (c) Total body water (kg), calculated from total body water (%) | Both sexes | All | RI, ht, BMI, sex | - | - | 0.691* | 0.964 |
| | | Boys | | | | 0.677 | 0.963 |
| | | Girls | | | | 0.704 | 0.961 |
| | Sex-specific | Boys | RI, ht, BMI | - | - | 0.735 | 0.957 |
| | | Girls | RI, ht, BMI | | | 0.732 | 0.958 |

*wt*, weight(kg); *ht*, height(cm); *R*, resistance at 50KHz; *RI*, height(cm)$^2$/R (resistance index); $r^2$, r squared calculated from multiple regression fit; *max(p)*, maximum of p-values in regression model; *RMSE(PE)*, root mean squared error of predictions as calculated from the out-of-sample predicted values (prediction error); $r^2(PE)$, proportion of variance explained by the out-of-sample predicted values; *%p*, percentage points

* best performing predictive equations for TBW(kg) and TBW(%)

**Table 4. Best predictive equations for TBW (kg) and TBW(%).**

| |
|---|
| (a) TBW (kg) = -3.5202 + 0.00950*R + 0.231103*RI + 0.27658*weight—0.9182*sex |
| (b) TBW (%) = 100*(1.11373 + 0.0037049*RI -0.25778*height—0.01812*BMI—0.02614*sex) |
| (c) TBW (kg) = weight*(1.11373 + 0.0037049*RI -0.25778*height—0.01812*BMI—0.02614*sex) |

$R$, resistance ($\Omega$) at 50 kHz; $RI$, resistance index, height$^2$/resistance (cm$^2$/$\Omega$), *TBW*, total body water
Weight is entered in kg, height in m, and sex coded: boys = 0, girls = 1.
Equation (c), which predicts TBW (kg) by multiplying TBW (%) by weight, has better predictive performance than (a), which models TBW (kg) directly.

values explain 63.8% of its variance (Table 3). Comparisons of out-of-sample predicted values with measured values and prediction errors, based on a 10-fold split of data, applying the RCV approach, are illustrated in Bland-Altman plots in Figs 2 and 3. For TBW (kg), there was a bias of 0.01 kg (95% limits of agreement -1.34, 1.36) for boys and -0.01 kg (-1.41, 1.38) for girls (Fig 2). For TBW (%), bias for boys was 0.0% (-4.2%, 4.2%) and for girls 0.0% (-3.7%, 3.7%) (Fig 3).

## Discussion

We derived prediction equations for total body water among Malawian 11-14-year-old children using Seca m515 8-polar bioimpedance device, with deuterium dilution technique as gold standard. The variables included in our equation, in particular the bioimpedance index [32] have also been found important in previous studies pertaining to various populations [8, 15–17, 33]. As far as we are aware this study is unique as we have carried out validation of BIA equations with deuterium dilution as the reference method among young teenage or early pubertal adolescents in Southeastern Africa. Similar studies focusing on young children have been carried out in Senegal [15], Nigeria [16] and the Gambia [17] for other bioimpedance devices.

Due to the lack of previous equations for this population and to the availability of a relatively large number of variables, we used a data-driven predictive modeling approach utilizing repeated 10-fold cross-validation, initially treating all variables in the data as potential predictors [27–29]. We acknowledge that predictive models should reflect existing theory and should utilise prior information whenever possible, and we find it reassuring that our proposed equations are not contradictory in this regard. Using the predictive modeling approach, we found an accurate parsimonious model with the best set of predictors selected from several anthropometric and bioimpedance measurements. The predictive approach applying repeated cross validation was used in a 2021 study in European children and adolescents from 5 countries [34] but otherwise appears to be surprisingly little used in previous bioimpedance equation model selection studies despite the fact that predictive performance should be the only true test of the resulting model [27].

Interestingly, we found that the prediction accuracy for TBW (kg) improved by first finding a linear regression equation for the body water percentage TBW (%) and then predicting TBW (kg) using that equation by multiplying the predictions for TBW (%) by person's weight. This can be explained by the large variation in body weight, and by the observation that if predictive models for TBW (kg) are estimated separately by body weight category, regression coefficients (apart from weight) increase in absolute value with weight, indicating interaction effects.

We evaluated prediction equations based on estimated out-of-sample prediction errors. In bioimpedance studies one part of the data is typically set aside as a test set representing yet-to-be-observed data. However, unless the sample size is very large, using a single split sample is known to be ineffective both for variable selection and for error estimation [35, 36]. Therefore,

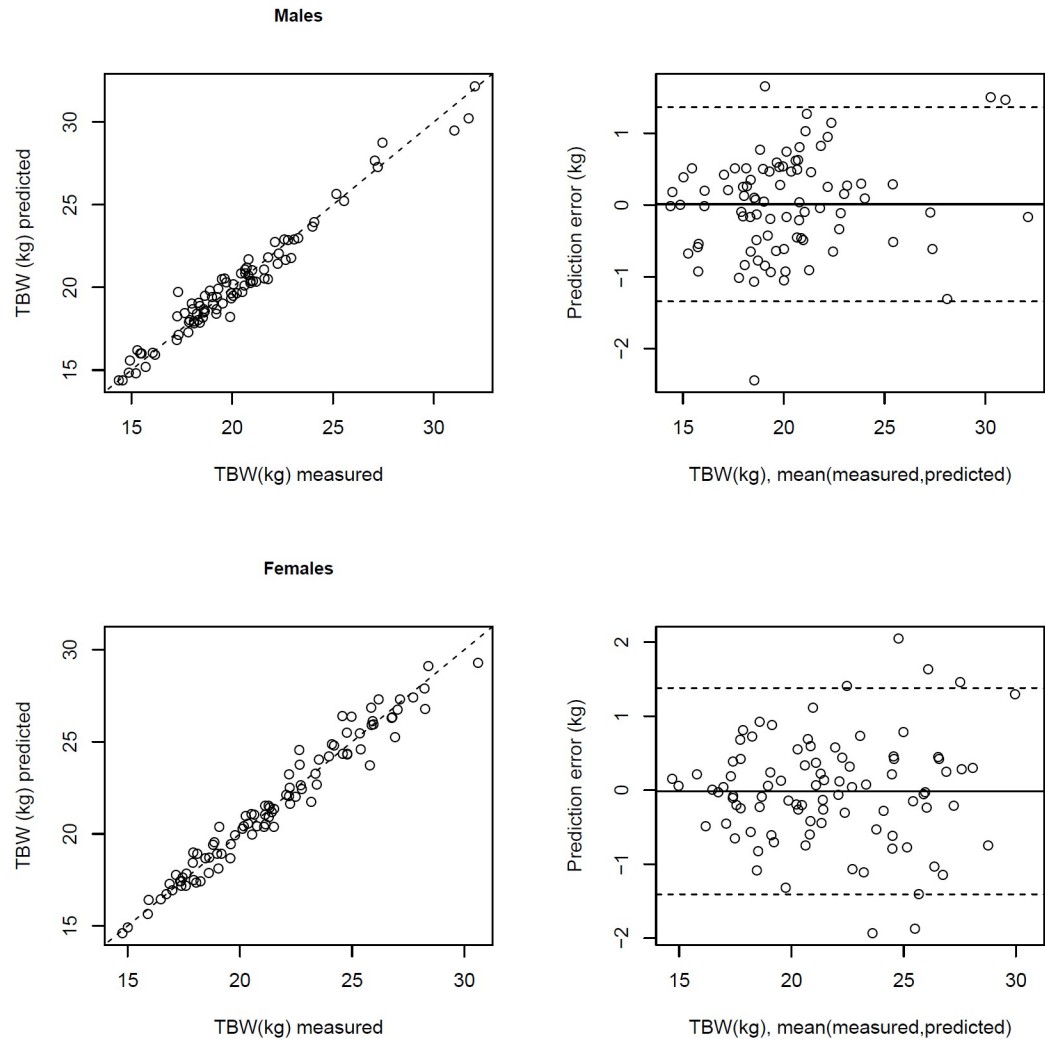

**Fig 2. Correlation between total body water (TBW) in kg measured by deuterium dilution with TBW (kg) predicted by bioimpedance using the predictive equation in Table 4 ($r^2$ = 0.96 for males, 0.96 for females), and Bland-Altman plots showing the difference in TBW (kg) measured by deuterium dilution and predicted by the bioimpedance equation against their mean.**

for increased accuracy, we used each fold in a 10-fold split in turn as a test set and thereby obtained out-of-sample predictions for each observation and, due to repeated splitting, obtained reliable out-of-sample prediction error estimates [30] enabling comparison of different modeling options by comparing their respective external prediction accuracies. We demonstrated that applying BIC and particularly AIC variable selection criteria in our application would likely have incorporated too many predictors in the model compared to applying 10-fold cross validation in variable selection. Importantly, the variable selection cross validation was nested within that used for the prediction error estimation [30] However, as the final model comparison among the different models in Table 3 utilised the model specific out-of-sample RMSE-values, the RMSE of the selected model is still likely to slightly underestimate the true prediction error.

Our mean TBW (kg) prediction error of 0.691 kg corresponds to approximately 0.2 SD and also compares favourably with that of 0.89 kg in a study in younger children in Senegal [15].

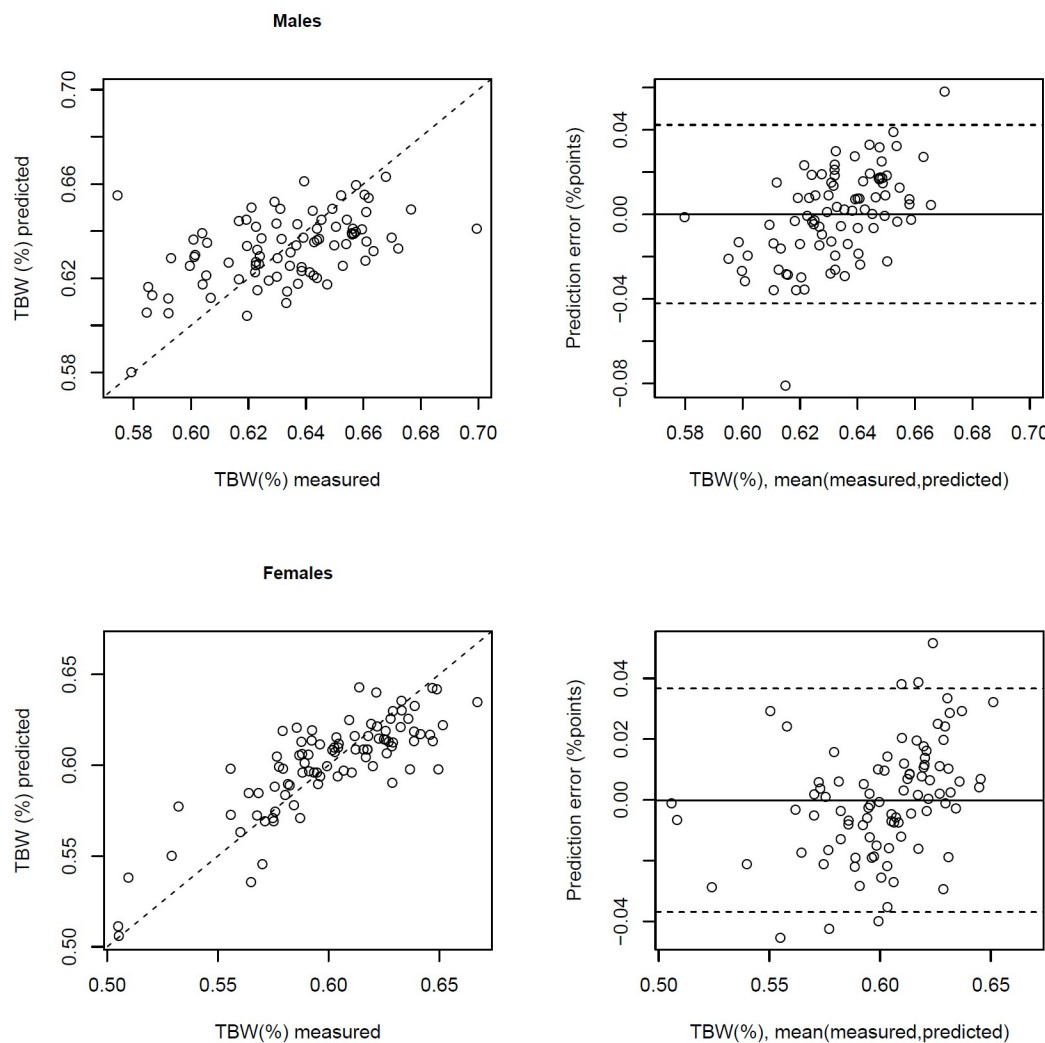

**Fig 3. Correlation between total body water (TBW) in % measured by deuterium dilution with TBW (%) predicted by bioimpedance using the predictive equation in Table 4 ($r^2$ = 0.26 for males, 0.68 for females), and Bland-Altman plots showing the difference in TBW (%) measured by deuterium dilution and predicted by the bioimpedance equation against their mean.**

The prediction equations contribute towards meeting the increasing need for affordable but accurate body composition measures for nutrition and health studies in populations of children and adolescents in rural Sub-Saharan Africa, which is essential to understand body composition and its changes in changing nutritional environments. As to predictive power on an individual level, the limits of agreement for TBW (kg) indicate that 95% of the adolescents would have a prediction error less than approximately 1.4 kg, corresponding to 0.4 SD, in either direction. This would set some limitations to the utility of BIA measurement in clinical practice and other contexts whether accurate measurements for all individuals would be of importance.

Our study has some limitations. An inherent limitation for studies creating and cross-validating prediction equations for body composition is that they are specific to the device and population. This is significant in particular in adolescence when puberty brings about rapid

changes in body composition. We included Tanner pubertal staging in the initial model, but it was not selected by the modelling procedure. However, only girls had significant variation in pubertal stage, whereas over 90% of boys were still prepubertal on Tanner stage 1. The timing of puberty is largely in accord with what has been observed in a previous study of 15-year-olds in the same rural area [37]. Boys experience pubertal growth spurt later than girls: their peak growth velocity takes place at Tanner stage 3–4, as compared with stage 2–3 in girls [38]. Consistent with this, height z-scores were lower in boys than in girls. As a further limitation, only a half or the participants had fasted overnight, and participants were not systematically asked to empty their bladder after arriving to the clinic, which could increase random error in the measurements. The Seca m515 device is most suitable in studies in a single clinic. The device is relatively costly and more difficult to transport than a number of portable devices in the market.

## Conclusions

We generated a prediction equation to convert raw bioimpedance data from Seca m515 bioimpedance analyser to total body water, from which other body composition measures can be calculated. These equations enable measurement of body composition by Seca m515 body composition analyser in prepubertal and pubertal adolescents in clinic-based population studies in rural Africa, where electricity is available.

## Supporting information

**S1 Fig. Scatterplots of total body water (TBW) against 5 anthropometric and 3 bioimpedance measurements for girls.** Estimated regression lines plotted. Correlation coefficient r (with p-value) listed on bottom left corner. R = resistance, Xc = reactance, RI = resistance index, TBW (kg) = total body water in kilograms, TBW(%) = total body water as percentage of weight.
(TIF)

**S2 Fig. Scatterplots of total body water (TBW) against 5 anthropometric and 3 bioimpedance measurements for boys.** Estimated regression lines plotted. Correlation coefficient r (with p-value) listed on bottom left corner. R = resistance, Xc = reactance, RI = resistance index, TBW (kg) = total body water in kilograms, TBW(%) = total body water as percentage of weight.
(TIF)

**S1 File. Inclusivity questionnaire.**
(DOCX)

**S2 File. R code for cross-validating the predictive equations.**
(R)

## Acknowledgments

We thank John Kamwendo for excellent technical assistance in the Fourier-transform infrared spectroscopy measurements and the research personnel of the Lungwena Research Centre for their efforts in participant recruitment and examination.

## Author Contributions

**Conceptualization:** Pieta Näsänen-Gilmore, Per Ashorn, Ulla Ashorn, Eero Kajantie.

**Data curation:** Pieta Näsänen-Gilmore, Markku Nurhonen, Lotta Hallamaa, Charles Mangani, Ulla Ashorn, Eero Kajantie.

**Formal analysis:** Markku Nurhonen.

**Funding acquisition:** Per Ashorn, Ulla Ashorn, Eero Kajantie.

**Investigation:** Chiza Kumwenda, Markku Nurhonen, Lotta Hallamaa, Charles Mangani, Per Ashorn, Ulla Ashorn, Eero Kajantie.

**Methodology:** Pieta Näsänen-Gilmore, Chiza Kumwenda, Markku Nurhonen, Per Ashorn, Ulla Ashorn, Eero Kajantie.

**Project administration:** Pieta Näsänen-Gilmore, Chiza Kumwenda, Lotta Hallamaa, Charles Mangani, Per Ashorn, Ulla Ashorn, Eero Kajantie.

**Resources:** Pieta Näsänen-Gilmore, Per Ashorn, Ulla Ashorn, Eero Kajantie.

**Supervision:** Pieta Näsänen-Gilmore, Ulla Ashorn, Eero Kajantie.

**Validation:** Chiza Kumwenda, Markku Nurhonen.

**Visualization:** Markku Nurhonen.

**Writing – original draft:** Pieta Näsänen-Gilmore, Eero Kajantie.

**Writing – review & editing:** Pieta Näsänen-Gilmore, Chiza Kumwenda, Markku Nurhonen, Lotta Hallamaa, Charles Mangani, Per Ashorn, Ulla Ashorn, Eero Kajantie.

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
