## [Decision Letter · Decision Letter 0]

1 Dec 2021

PONE-D-21-31743Body composition among Malawian young adolescents: Cross-validating predictive equations for bioelectric impedance analysis using deuterium dilution methodPLOS ONE

Dear Dr. Kajantie,

Thank you for submitting your manuscript to PLOS ONE. After careful consideration, we feel that it has merit but does not fully meet PLOS ONE’s publication criteria as it currently stands. Therefore, we invite you to submit a revised version of the manuscript that addresses the points raised during the review process.

Your manuscript has now been reviewed by two experts in the field. While both reviewers agreed that the study is of intrest for the scientific community, some major concerns however have been raised and will therefore need to be adressed before any final decision can be reached on the manuscript. Below please find the specific comments and suggestions brought by the two experts. Among those, please take special attention on (i) the data processing that should be expanded and appropriate analyses conducted, and (ii) the existence of major weaknesses in the protocol and analysis requiring the addition of some complementary information.

We look forward to receiving your revised manuscript.

Kind regards,

Sylvain Giroud, PhD

Academic Editor

PLOS ONE

Journal Requirements:

3. Please include a complete copy of PLOS’ questionnaire on inclusivity in global research in your revised manuscript. Our policy for research in this area aims to improve transparency in the reporting of research performed outside of researchers’ own country or community. The policy applies to researchers who have travelled to a different country to conduct research, research with Indigenous populations or their lands, and research on cultural artefacts. The questionnaire can also be requested at the journal’s discretion for any other submissions, even if these conditions are not met.  Please find more information on the policy and a link to download a blank copy of the questionnaire here: https://journals.plos.org/plosone/s/best-practices-in-research-reporting. Please upload a completed version of your questionnaire as Supporting Information when you resubmit your manuscript.

 [The original LAIS study was supported by grants from the Academy of Finland (grants 79787 and 207010 to PA; aka.fi), the Foundation of Pediatric Research in Finland (lastentautientutkimussaatio.fi ; PA), and the Medical Research Fund of Tampere University Hospital (PA). This study was supported by the Academy of Finland (EK; aka.fi), Finnish Medical Foundation (Finska Läkaresällskapet) (EK), Foundation for Pediatric Research in Finland (EK, lastentautientutkimussaatio.fi), Novo Nordisk Foundation (EK, novonordiskfonden.dk), Signe and Ane Gyllenberg Foundation (PNG, EK; gyllenbergs.fi) and Sigrid Juselius Foundation (EK; sigridjuselius.fi).]

7. Please upload a copy of Figures 2 and 3, to which you refer in your text on page 12. If the figure is no longer to be included as part of the submission please remove all reference to it within the text.

Reviewers' comments:

Reviewer's Responses to Questions

**Comments to the Author**

1. Is the manuscript technically sound, and do the data support the conclusions?

Reviewer #1: Partly

Reviewer #2: Yes

2. Has the statistical analysis been performed appropriately and rigorously? 

Reviewer #1: No

Reviewer #2: Yes

3. Have the authors made all data underlying the findings in their manuscript fully available?

Reviewer #1: Yes

Reviewer #2: Yes

4. Is the manuscript presented in an intelligible fashion and written in standard English?

Reviewer #1: Yes

Reviewer #2: Yes

5. Review Comments to the Author

Reviewer #1: A generally clearly written manuscript that describes an ostensibly well-conducted study. Some points for the authors to consider.

Line 66 onward. This is a simplistic description of BIA but not inaccurate. The authors might consider describing the differences between single, multi-frequency and BIS. I note that they used what is essentially a BIS device (19 frequencies) but BIS analysis was not used, preferring empirical equation generation. Why not? This could be readily performed since the authors state that all raw data were obtained (line 127). The authors could then test whether their equations perform better or worse than BIS prediction (see publications by Lyons-Reid for further information - in this regard some of the citing references are rather old and more up-to-date reference could be used).

Line 68 "Resistance and reactance depend on water content of the tissue and, with appropriate prediction equations," is not quite correct. Reactance Xc is determined by the capacitive nature of cell membranes. BIS does not use empirical predictive equations.

Line 73 onward. I agree that DD is a reference method for TBW but do not agree that it is difficult in children and adolescents. There are many studies attesting to its use in this population and, indeed, the IAEA, whose protocols were followed promote and support its use in children. Furthermore, if absolutely necessary fluid intake is allowed and can be corrected for (see Schoeller's chapter on "Hydrometry" in "Human Body Composition" by Heymsfield.

Line 79. There may not be other studies validating using DD but there are papers that use DXA which is again an accepted reference method , e.g. van Zyl et al. I suggest that you make this point.

Lines 82, 92 and elsewhere please check for inappropriate capitalization.

Line 124. BIA is a technique that requires careful standardisation of the protocol. Please provide full details here. See papers by Brantlov for guidance on reporting standards.

The DD method is appropriately described but details of quality control procedures are required as discussed in the IAEA monographs on the technique.

Data analysis The initial regression strategy seems to be rather shotgun - 23 variables! For example, why resistance at 5 frequencies but Xc at only 50 kHz (note kHz NOT KHz)? Why these particular frequencies? What is the rationale for including R and RI in the prediction equation 1 in Table 4? BIA theory clearly shows that conductive volume (e.g. TBW) = rho H^2/R NOT simply R. R alone is going to be raled to length and cross-section of the conductive volume which relates to TBW but will not be independent of RI. Simply using VIF to eliminate (choose) variables is inappropriate, select on strong theoretical underpinnings.

Line 189. I do not understand this. Magnitude of R is clearly frequency dependent, the basic BIA Cole model shows this and it is readily observed in practice - look at your R data for the 19 frequencies. I do not see the theoretical rationale for using Xc at different frequencies. This is getting close to a BIS-style frequency-dependent analysis but does not appear to be justified here. This is important since it gets to the heart of your equation generation and needs better justification.

Line 194 onward. The K-fold validation process is OK to get an idea of overall model error but inevitably the mean values are equivalent to the overall population mean, hence you essentially 0 biases seen in the LOA analysis Line 272 on. The "true bias" should be assessed by using an independent cross-validation group. This can be undertaken with your data by splitting the group into two sex-stratified cohorts of say 2/3 and 1/3. Generate the equation in the 2/3 group and test in the 1/3 using LOA and correlation (use concordance not Pearson). This should be done, at present your bias values are meaningless.

Line 212 and equation b. Given the relationship between RI and TBW above, equations should not be generated in relative (%) terms. Delete this.

Results are generally well presented with the caveats noted above. But the LOA are not reflective of "true" LOA; the same sample has been used to generate the equation and to test the equation - this is circular. The split sample test described above should be used. The "true" LOA are likely to be somewhat larger. For the LOA as given this was around +/- 1.4 kg. Since TBW was around 20 kg this means that the LOA are around +/- 7% Is this clinically acceptable? See Ward in EJCN for a discussion on this point. This point needs to be discussed as well as the overall error (lines 304 on) since it represents the predictive power for an individual rather than for a population (the bias and associated error).

Discussion

Line 288 on the authors appear to rely on the statistical approach to parameter selection. This is important but is not the most important. Parameters should only be included on justifiable biological model grounds not simply BIC or AIC criteria. I repeat that the authors need to justify physiologically and congruent with the BIA model the choice of predictive parameters.

Please provide the LOA plots as supplementary data (after the re-analysis as requested).

Line 299 This needs to address the point made above that K-folding does not highlight true predictive error for individuals and does not provide a true bias estimate.

The seca device used here provides its own predictions of TBW. How do these compare to the the DD and the predictive equations output? It is important that this be tested along with some similar published equations for cognate groups e.g. the refs cited by the authors. This is important since as the authors allude equations are population specific. Also they should test against the fundamental BIS approach which is purported to not be pop-specific.

In conclusion, the study data collection appears sound but the data analysis should be expanded and appropriate analyses conducted.

Reviewer #2: Comments to the authors

In this study, the authors derived a predictive equation to estimate fat-free mass from total body water measured by BIA in children (11-14 years) in Malawi using isotopic dilution as the gold standard method. We are lacking inexpensive, easy to implement and reliable methods to assess body composition in African populations, especially in children. This study is therefore important as it is filling a gap in knowledge and will allow to conduct further nutrition studies in these vulnerable populations that require specific public health guidelines and actions.

Although we appreciate the goal of the study, the current scientific approach presents few weaknesses that need to be addressed. Below we are suggesting some edits and alternative approaches to the authors hoping they will allow to improve the quality of the manuscript.

Major comments

It is unclear why the predictive equations, RMSE, p-values etc are different for TBW expressed in % and in kg, when one is the direct relationship of the other (Table 3). The methods/models used to derive TBW in kg and TBW in kg calculated from TBW in % is also confusing. Please further explain your approach and justify why those equations are different.

As mentioned by the authors, the fact that all the children were not fasted and had an empty bladder is major limitation as it is not following the standard protocol for these measurements. Please consider running the analysis in the children who were both fasted and with an empty bladder and then compare with your current equations. If the two are close, please provide the equations in the subgroup as supplemental material. If they are different, please only provide the equation derived with the kids who complied with the instructions.

We suggest to only use height and weight in the equations and not BMI as it is a ratio of the two first variables.

Please consider discussing the variance, mean bias and mean prediction error of your model in light of the clinically significant differences/changes for fat-free mass in children to know whether or not this method could be powerful enough in a clinical setting.

Although we understand that the BIA from Semca is likely more cost-effective than the use of deuterated water because of the cost of the samples analysis, it is not necessarily an inexpensive method. The purchase of this device cannot be afforded by all medical settings and its access may likely be restricted major clinics. It would have been interesting to also compare those two methods to another cheap method such as the skinfold thickness method. Even if much more inaccurate, such a method could be interesting for large epidemiological studies, especially in ultra-rural and remoted areas. Other devices such as portable Tanita BIS could also be easier to use in such environments. Although we understand the authors derived an equation using a device they had access to, it does not necessarily address the need for a method to estimate body composition that is easy to implement, cheap and accurate. We would like to invite the authors to discuss those considerations.

Please clearly specify the inclusion/exclusion criteria of the study participants.

Please proof-read English.

Introduction

Lines 61-62 : BMI being used as an index of overweight and obesity, it is logical that BMI increases along with the obesity epidemic progresses. Please rephrase this sentence.

Line 62: Obesity is also due to malnutrition. Did you mean “undernutrition?

Line 65: Consider adding a sentence on the current needs, i.e. developing an accurate, simple, cheap/affordable and easy to administer method that can be used to assess body composition in large population studies to better delineate future public health guidelines.

Line 71: Remove “a” prior to “healthy adults”

Line 72: Replace total body weight by total body mass.

Line 73-75: This method is also pricy, requires specific devices (IRMS or infrared spectroscopy) for samples analysis that are possessed by only few labs around the world, that are very expensive and need specific expertise.

Line 78-79: The gap in knowledge is here implicit but needs to be clearly described like it is nicely done in the abstract.

Line 81: It may not be needed to provide the details of the BIA device in the introduction section.

The scientific premise of the study would benefit for being a bit more substantial; for example, it is also important to justify the study population, children and young adolescents. Why an age range comprised between 11 and 14 years old? We also invite the authors to explain why they believe BIA is more adequate than other available methods for measuring their body composition (e.g., skinfold thickness, BIS, others).

Line 86: Were the informed consent signed or orally obtained? In a population where a large portion of the population may be illiterate, it is important to provide more information about the methods for consent obtention.

M&M

Line 95: Sample size of mothers? Sample size of the kids who were potentially available for this study?

Line 118: How many kids reported not being fasted on the study day?

Lines 120-121: It is not clear why it was not asked to all kids to empty their bladder prior to dosing.

Body composition by bioimpedance: please provide more information about the measurement including data analysis to obtain body composition, placement of the electrodes, etc.

It would have been interesting to take the measurement in duplicate.

It is unclear why weight was measured at the beginning and at the end of the experiment, and why was it taken into account? To what the weight change over few hours can be attributed? Can it be attributed to the intra-individual precision of the BIA? This information on the device should be provided.

Body composition by deuterium dilution

Line 151: Add “post-dose” after “the 3 hour and 4 hour”

Please add a reference showing that the method using saliva samples has been validated against the one with urinary samples.

What methods/approach did you use to limit isotopic fragmentation when collecting saliva samples, especially in a likely hot environment?

Were the analysis of the samples calibrated against standards, e.g. SMOW2 SAP?

Line 170: A brief description of the method would be welcomed.

Line 174: Please ad that FM was obtained by subtracting FFM to BM.

Line 177: Please rephrase the first sentence. (“used” instead of “applied”?)

Please indicate how many of the 240 kids reported to the study visit following your invitation.

Provide sample size for each dichotomous variable.

Line 185: how was multicollinearity tested for?

Lines 188-189: Please rephrase this sentence.

Lines 191-192: please rephrase by avoiding future tense.

Line 243-244: Why did you decide to only use the RCV predictors? How many equations did you end up getting in total? What was the range of the percentage of the variance that was obtained with those equations?

Line 217: Please spell out “supplemental”.

Results

Line 222: Please rephrase the first sentence by modifying “entered”.

Table 2: Add data for fat-free mass and body mass.

In Table 3, it is unclear how TBW in kg was estimated on the first line.

Table 4: It is unclear what (a), (b) and (c) are referring to, the three lines/models in Table 3?

It is unclear why the predictions are worse for TBW in % than in kg when the only difference is the consideration of body mass that was an independent measure.

Figures 2 and 3 are missing.

Discussion

It is difficult to know how generalizable the results of this study are and if this equation can be used for populations other than Malawi kids. Would the authors have possibilities to test the equation of other data sets? We encourage you to further compare your results with those obtained in populations from Senegal, Gambia and Nigeria.

It would be important to discuss your results on FFM and FM against the clinically significantly difference in children. The bias interval of confidence for TBW being large (+/-4.2% for boys; 3.7% for girls), we can wonder if the BIA method would allow to detect any significant differences across populations or changes.

The need for electricity on the field in this population maybe a limitation. One can wonder if the skinfold thickness method, albeit less accurate, not be more appropriate for this population.

6. PLOS authors have the option to publish the peer review history of their article (what does this mean?). If published, this will include your full peer review and any attached files.

Reviewer #1: **Yes: **Leigh Ward

Reviewer #2: **Yes: **Audrey Bergouignan

---

## [Author Response · Author response to Decision Letter 0]

26 May 2022

Please see the response letter included in the submission.

---

## [Decision Letter · Decision Letter 1]

17 Aug 2022

PONE-D-21-31743R1Body composition among Malawian young adolescents: Cross-validating predictive equations for bioelectric impedance analysis using deuterium dilution methodPLOS ONE

Dear Dr. Kajantie,

Thank you for submitting your manuscript to PLOS ONE. After careful consideration, we feel that it has merit but does not fully meet PLOS ONE’s publication criteria as it currently stands. Therefore, we invite you to submit a revised version of the manuscript that addresses the points raised during the review process. In particular, please take a special attention at the -although minor- important comments of reviewer-2 who pointed again some issues that were raised by both reviewers during the first round of review. This new step of review requires additonal corrections and justifications of some aspects of your study. Please address these thourougly this time and revised your mansucript accordingly.

We look forward to receiving your revised manuscript.

Kind regards,

Sylvain Giroud, PhD

Academic Editor

PLOS ONE

Journal Requirements:

Reviewers' comments:

Reviewer's Responses to Questions

**Comments to the Author**

1. If the authors have adequately addressed your comments raised in a previous round of review and you feel that this manuscript is now acceptable for publication, you may indicate that here to bypass the “Comments to the Author” section, enter your conflict of interest statement in the “Confidential to Editor” section, and submit your "Accept" recommendation.

Reviewer #1: All comments have been addressed

Reviewer #2: (No Response)

2. Is the manuscript technically sound, and do the data support the conclusions?

Reviewer #1: Yes

Reviewer #2: Yes

3. Has the statistical analysis been performed appropriately and rigorously? 

Reviewer #1: Yes

Reviewer #2: Yes

4. Have the authors made all data underlying the findings in their manuscript fully available?

Reviewer #1: Yes

Reviewer #2: No

5. Is the manuscript presented in an intelligible fashion and written in standard English?

Reviewer #1: Yes

Reviewer #2: Yes

6. Review Comments to the Author

Reviewer #1: The authors have largely satisfactorily addressed the issues raised in my original review and have modified the manuscript accordingly.

However, I note that both I and the second reviewer raised the issue of assessing multicollinearity in the regression. The authors have chosen not to pursue this. I am not going to insist upon this point but I am not entirely convinced by their arguments that this should be ignored.

Similarly, I am not entirely convinced of the inadequacy of a single split-plot approach to agreement assessment. This approach replicates "real world" measurement where single application of a prediction method in essence reflects a single sample approach and the errors inherent therein. The multi-fold approach as acknowledged by the authors loses knowledge of this variance. But again, opinions vary in statistics as elsewhere!

Reviewer #2: Comments to the authors

The authors have provided with extensive answers to each comment and they made an effort to justify their positions and choices.

However, I still have few minor comments prior to consider this paper ready for publication though. The goal is to improve the quality of the manuscript.

- Line 241-243: please clarify the methods.

- Line 318-321: Please add p-value and R2 for the associations.

- It is still unclear to me why you could not provide the equations with the 52 subjects who had both empty bladder and who were fasted. This could be shown as supplemental material.

- I am not convinced by the arguments of your choice of height and BMI for the equations. R1 asked to verify collinearity using VIF. You don’t seem to have run this test. I however agree if would be important to do and thus check if your choice of height and BMI is appropriate using a statistical test.

- In the discussion, conclusion and abstract, it needs to be much clearer that this device is appropriate to be used at group/pop level and not at the individual level. Although it has been mentioned, it is still not a clear statement. I would also suggest to run a sample size power analysis using FFM to estimate the sample size needed for such a study in children/adolescents. This will give the reader an idea of how big the study would need to be.

- Line 398-402: Please consider adding the fact that the device is expensive and not transportable which will limit studies especially on the field.

- In the introduction, the choice of BIA over other available methods is still not clear to me.

- I still don’t understand the weight change during the isotopic dilution measurement. What is the reason for this change? In your answer, you stated that weight change was included to estimate any possible isotopic fragmentation. While this is interesting, it seems it would be possible only if you knew the exact value of TBW, which you don’t especially that you assumed (like any other investigator using this method) that you collected urines at the plateau. So I am still having hard time getting the point here.

7. PLOS authors have the option to publish the peer review history of their article (what does this mean?). If published, this will include your full peer review and any attached files.

Reviewer #1: **Yes: **Leigh Ward

Reviewer #2: **Yes: **Audrey Bergouignan

---

## [Author Response · Author response to Decision Letter 1]

19 Mar 2023

Please see attached separate point-by-point response to the Editor's and Reviewers' comments.

---

## [Decision Letter · Decision Letter 2]

27 Mar 2023

Body composition among Malawian young adolescents: Cross-validating predictive equations for bioelectric impedance analysis using deuterium dilution method

PONE-D-21-31743R2

Dear Dr. Kajantie,

We’re pleased to inform you that your manuscript has been judged scientifically suitable for publication and will be formally accepted for publication once it meets all outstanding technical requirements.

Kind regards,

Sylvain Giroud, PhD

Academic Editor

PLOS ONE

Reviewers' comments:

Reviewer's Responses to Questions

**Comments to the Author**

1. If the authors have adequately addressed your comments raised in a previous round of review and you feel that this manuscript is now acceptable for publication, you may indicate that here to bypass the “Comments to the Author” section, enter your conflict of interest statement in the “Confidential to Editor” section, and submit your "Accept" recommendation.

Reviewer #2: All comments have been addressed

2. Is the manuscript technically sound, and do the data support the conclusions?

Reviewer #2: Yes

3. Has the statistical analysis been performed appropriately and rigorously? 

Reviewer #2: Yes

4. Have the authors made all data underlying the findings in their manuscript fully available?

Reviewer #2: Yes

5. Is the manuscript presented in an intelligible fashion and written in standard English?

Reviewer #2: Yes

6. Review Comments to the Author

Reviewer #2: Thank you for addressing all the comments. Thank you.

7. PLOS authors have the option to publish the peer review history of their article (what does this mean?). If published, this will include your full peer review and any attached files.

Reviewer #2: **Yes: **Audrey Bergouignan

---

## [Editor Report · Acceptance letter]

3 Apr 2023

PONE-D-21-31743R2 

Body composition among Malawian young adolescents: Cross-validating predictive equations for bioelectric impedance analysis using deuterium dilution method 

Dear Dr. Kajantie:

I'm pleased to inform you that your manuscript has been deemed suitable for publication in PLOS ONE. Congratulations! Your manuscript is now with our production department. 

Kind regards, 

on behalf of

Dr. Sylvain Giroud 

Academic Editor

PLOS ONE